

# Design, synthesis and antimicrobial activities of novel 1,3,5-thiadiazine-2-thione derivatives containing a 1,3,4-thiadiazole group

Jinghua Yan[1], Weijie Si[1,2], Haoran Hu[1], Xu Zhao[1], Min Chen[1] and Xiaobin Wang[1]

[1] Jiangsu Key Laboratory of Pesticide Science, College of Sciences, Nanjing Agricultural University, Nanjing, Jiangsu Province, China
[2] Key Laboratory of Monitoring and Management of Crop Diseases and Pest Insects, Ministry of Agriculture, Nanjing Agricultural University, Nanjing, Jiangsu Province, China

## ABSTRACT

A series of novel 1,3,5-thiadiazine-2-thione derivatives containing a 1,3,4-thiadiazole group was designed and synthesized. The structures of all the compounds were well characterized using $^1$H NMR, $^{13}$C NMR and high-resolution mass spectrometer, and further confirmed by the X-ray diffraction analysis of **8d**. The antimicrobial activities of all the target compounds against *Xanthomonas oryzae pv. oryzicola*, *X. oryzae pv. oryzae*, *Rhizoctonia solani* and *Fusarium graminearum* were evaluated. The in vitro antimicrobial bioassays indicated that some title compounds exhibited noteworthy antimicrobial effects against the above strains. Notably, the compound *N*-(5-(ethylthio)-1,3,4-thiadiazol-2-yl)-2-(5-methyl-6-thioxo-1,3,5-thiadiazinan-3-yl)acetamide (**8a**) displayed obvious antibacterial effects against *X. oryzae pv. oryzicola* and *X. oryzae pv. oryzae* at 100 μg/mL with the inhibition rates of 30% and 56%, respectively, which was better than the commercial bactericide thiodiazole-copper. In addition, the anti-*R. solani* $EC_{50}$ value of **8a** was 33.70 μg/mL, which was more effective than that of the commercial fungicide hymexazol (67.10 μg/mL). It was found that the substitutes in the 1,3,5-thiadiazine-2-thione and the 1,3,4-thiadiazole rings played a vital role in the antimicrobial activities of the title compounds. More active title compounds against phytopathogenic microorganisms might be obtained via further structural modification.

## INTRODUCTION

A variety of plant diseases, caused by pathogenic organisms, seriously affects the crop production, leading to tremendous losses to the agricultural economy every year (*Wilson & Talbot, 2009*; *Liu et al., 2013*). Besides, the rapid emergence of resistant strains against traditional antimicrobial agents has become a huge challenge in the agricultural industry (*Wang et al., 2013*). In the past decades, researchers have found a large number of bioactive molecules with strong inhibitory effects on phytopathogenic bacteria and fungi. However,

Corresponding author
Min Chen, chenmin@njau.edu.cn

these compounds are rarely used in crop production due to the structural instability or poor control in farmland. Therefore, it is tardy to search highly-effective and eco-friendly agrochemicals for fighting against agricultural pathogenic microorganisms (*Qian, Lee & Cao, 2010*; *Li et al., 2018*).

1,3,5-Thiadiazine-2-thione derivatives were attractive bioactive molecules and exhibited antibacterial (*Mao et al., 2017*), antifungal (*Vicentini et al., 2002*), herbicidal (*Vicentini et al., 2005*), anticancer (*El-Shorbagi et al., 2018*), antileishmanial (*Arshad et al., 2018*), antiepileptic (*Semreen et al., 2010*), antimalarial (*Coro et al., 2006*), antioxidant (*Ji et al., 2004*), antitubercular (*Katiyar et al., 2003*) and trypanocidal (*Coro et al., 2005*) activities. Notably, the agricultural application of 1,3,5-thiadiazine-2-thione derivatives has also attracted great attention by chemists and biologists in the last three decades. For example, dazomet (Fig. 1A) and milneb (Fig. 1B), containing the 1,3,5-thiadiazine-2-thione moiety, were developed as the important agricultural nematicide and fungicide, respectively (*Lam et al., 1993*; *Nakamura et al., 2010*). Recently, *Mao et al. (2017)* found that dazomet could be applied as a promising agricultural bactericide to effectively control ginger blast in field trials (*Mao et al., 2017*). Meanwhile, our previous work found that 1,3,5-thiadiazine-2-thione derivatives with an acylhydrazine group displayed obvious antifungal activity in vitro and in vivo (*Wang et al., 2018*).

1,3,4-Thiadiazole derivatives attracted great attention from biochemists due to their various bioactivities including antibacterial (*Zhong et al., 2017*), antifungal (*Chen, Li & Han, 2000*), insecticidal (*Luo & Yang, 2007*), antiviral (*Chen et al., 2010*), herbicidal (*Cummings, 2009*), anticancer (*Casey et al., 2004*), anti-tubercular (*Foroumadi, Kiani & Soltani, 2003*), antiparasitic (*Coura & De Castro, 2002*), antidepressant (*Siddiqui et al., 2011*), antioxidant (*Khan et al., 2010*) and anti-inflammatory (*Kumar et al., 2008*) activities. Among the above biological activities, the remarkable antimicrobial activity of 1,3,4-thiadiazole derivatives were well reported during the past decades. Thiodiazole-copper (Fig. 1C) and bismerthiazol (Fig. 1D), the representative agrochemicals containing the 1,3,4-thiadiazole group, were widely used to control bacterial diseases in crops. In addition, researchers found that the 1,3,4-thiadiazole derivatives bearing sulfides moiety could effectively inhibit various agricultural bacteria (*Wan et al., 2018*).

On the basis of the above analysis, a series of 1,3,5-thiadiazine-2-thione derivatives, containing 1,3,4-thiadiazole scaffold (Fig. 2), were obtained to find novel antimicrobial candidates. The 1,3,4-thiadiazole fragment was introduced into the 5-position of the 1,3,5-thiadiazine-2-thione according to the "combinatorial optimization" method (Fig. 3). The antibacterial activities against *Xanthomonas oryzae pv. oryzae* and *X. oryzae pv. oryzicola* and the antifungal activity against *Rhizoctonia solani* and *Fusarium graminearum* were evaluated. Furthermore, the preliminary biological assay showed that some of the title compounds exhibited good antibacterial and antifungal activities. To the best of our knowledge, this is the first report about the synthesis and antimicrobial activity of 1,3,5-thiadiazine-2-thione derivatives containing a 1,3,4-thiadiazole moiety.

**Figure 1 Bioactive compounds containing a 1,3,5-thiadiazine-2-thione or 1,3,4-thiadiazole fragment.**

**1a, 2a, 3a, 5a-5d**: R¹=Ph
**1b, 2b, 3b, 6a-6d**: R¹=Bn
**1c, 2c, 3c, 7a-7d**: R¹=4-FPh
**1d, 2d, 3d, 8a-8d**: R¹=Me

**4a, 5a, 6a, 7a, 8a**: R²=Me
**4b, 5b, 6b, 7b, 8b**: R²=Ph
**4c, 5c, 6c, 7c, 8c**: R²=4-MePh
**4d, 5d, 6d, 7d, 8d**: R²=4-ClPh

**Figure 2 Synthetic route to title compounds 5–8.**

**Figure 3 Design strategies for title compounds.**

## MATERIALS AND METHODS

### Materials

All the solvents and reagents were purchased from commercial suppliers and used without further purification. The reaction was monitored by thin layer chromatography on $GF_{254}$ silica gel plates, which was visualized at UV 254 nm. The melting points were measured with a SMP50 automatic melting point apparatus (Cole-Parmer, Staffordshire, England).

$^1$H and $^{13}$C NMR spectra were recorded on a BRUKER, AVANCE III 400 MHz and probe of BBO spectrometer (Bruker Corporation, Bremen, Germany) at room temperature with DMSO-$d_6$ as a solvent and TMS as an internal standard. High-resolution mass spectrometer (HRMS) data were measured on an AB SCIEX TripleTOF 5600$^+$ spectrometer (AB SCIEX, Fremingham, America) with the ESI charge source.

### General procedures for substituted 2-(6-thioxo-1,3,5-thiadiazinan-3-yl) acetic acids 3

Carbon disulfide (7.61 g, 100 mmol) was added dropwise to 100 mL 18% aqueous potassium hydroxide solution (18 g of potassium hydroxide dissolved in 82 g of water) containing phenylamine (9.31 g, 100 mmol). After being stirred for 4 h at room temperature, the reaction mixture turned from colorless to orange and appeared white solid (compound **1a**). Then, 37% formaldehyde solution (18.66 g, 230 mmol) was added into the reaction mixture, and the solution was stirred for another 1 h at room temperature. After being filtered, the filtrate was slowly dropped into a phosphate buffer solution (pH 7.8, 100 mL) containing glycine (7.51 g, 100 mmol). After being stirred for 2 h at room temperature and being filtered, the filtrate was washed with diethyl ether until the color of organic phase was changed to colorless. The water phase was acidified with dilute hydrochloric acid to generate white precipitates, at 0–5 °C. The precipitates were filtered, washed with iced ethanol, and dried to acquire the key intermediate **3a**. This method was suitable for the synthesis of compounds **3b–3d**. The yields of compounds **3a–3d** were between 46% and 71%.

### General procedures for substituted 5-(ethylthio)-1,3,4-thiadiazol-2-amines 4

Take the synthesis of compound **4a** as an example: 5-amino-1,3,4-thiadiazole-2-thiol (2.0 g, 15 mmol) and $K_2CO_3$ (2.76 g, 20 mmol) were dissolved in 20 mL DMF and stirred for 15 min at room temperature. The ethyl bromide (2.18 g, 20 mmol) was added dropwise into the above mixture and stirred for 8 h at room temperature. Then the mixture was poured into cold water (10 mL). The precipitated solid was filtered and recrystallized with the solvent of ethanol and water to gain pale yellow solid **4a**. The method was suitable for the synthesis of compounds **4b–4d**. The yields of compounds **4a–4d** were between 77% and 81%.

### General synthetic procedure for title compounds 5–8

The intermediate **3a** (0.50 g, 1.86 mmol), O-(Benzotriazole-1-yl)-N,N,N',N'-tetramethyluronium tetrafluoroborate (TBTU, 0.72 g, 2.24 mmol), triethylamine (0.38 g, 3.76 mmol) were added into dichloromethane (30 mL) and the mixture was stirred for 0.5 h at room temperature. Then, the intermediate **4a** (0.45 g, 2.80 mmol) was added and stirred for another 2 h at room temperature. The resulting precipitate was filtered, washed with dichloromethane, and dried to give the desired product **5a**. This method was suitable for the synthesis of title compounds **5–8** (*Si et al., 2019*). The yields of compounds **5–8** were between 41% and 79%.

### N-(5-(ethylthio)-1,3,4-thiadiazol-2-yl)-2-(5-phenyl-6-thioxo-1,3,5-thiadiazinan-3-yl)acetamide (5a)

White solid, m.p.158–160 °C, yield 61%; $^1$H NMR (400 MHz, DMSO-$d_6$) δ 12.73 (s, 1H, CONH), 7.47 (t, $J$ = 7.6 Hz, 2H, PhH), 7.35 (t, $J$ = 7.4 Hz, 1H, PhH), 7.25 (d, $J$ = 7.5 Hz, 2H, PhH), 4.76 (s, 2H, NCH$_2$N), 4.69 (s, 2H, SCH$_2$N), 4.02 (s, 2H, COCH$_2$), 3.23 (q, $J$ = 7.3 Hz, 2H, CH$_2$CH$_3$), 1.33 (t, $J$ = 7.3 Hz, 3H, CH$_3$); $^{13}$C NMR (101 MHz, DMSO-$d_6$) δ 193.4, 168.4, 159.3, 158.7, 130.0, 128.3, 127.7, 74.1, 59.3, 53.2, 28.5, 15.2; HRMS (ESI) $m/z$ calcd for C$_{15}$H$_{18}$N$_5$OS$_4$ ([M+H]$^+$): 412.0388, found: 412.0388.

### N-(5-(benzylthio)-1,3,4-thiadiazol-2-yl)-2-(5-phenyl-6-thioxo-1,3,5-thiadiazinan-3-yl)acetamide (5b)

White solid, m.p.175–176 °C, yield 66%; $^1$H NMR (400 MHz, DMSO-$d_6$) δ 7.46 (t, $J$ = 7.7 Hz, 2H, PhH), 7.40 (d, $J$ = 7.3 Hz, 2H, PhH), 7.34 (dd, $J$ = 17.5, 7.6 Hz, 3H, PhH), 7.28 (d, $J$ = 7.1 Hz, 1H, PhH), 7.24 (d, $J$ = 7.6 Hz, 2H, PhH), 4.75 (s, 2H, NCH$_2$N), 4.68 (s, 2H, SCH$_2$N), 4.48 (s, 2H, SCH$_2$Ph), 4.00 (s, 2H, COCH$_2$); $^{13}$C NMR (101 MHz, DMSO-$d_6$) δ 193.4, 168.4, 159.1, 158.7, 144.7, 137.2, 129.9, 129.5, 129.0, 128.3, 128.1, 127.7, 74.1, 59.3, 53.2, 38.0; HRMS (ESI) $m/z$ calcd for C$_{20}$H$_{19}$N$_5$NaOS$_4$ ([M+Na]$^+$): 496.0365, found: 496.0366.

### N-(5-((4-methylbenzyl)thio)-1,3,4-thiadiazol-2-yl)-2-(5-phenyl-6-thioxo-1,3,5-thiadiazinan-3-yl)acetamide (5c)

White solid, m.p.169–171 °C, yield 67%; $^1$H NMR (400 MHz, DMSO-$d_6$) δ 7.46 (t, $J$ = 7.2 Hz, 2H, PhH), 7.38–7.32 (m, 1H, PhH), 7.26 (dd, $J$ = 14.8, 7.7 Hz, 4H, PhH), 7.13 (d, $J$ = 7.5 Hz, 2H, PhH), 4.75 (s, 2H, NCH$_2$N), 4.68 (s, 2H, SCH$_2$N), 4.44 (s, 2H, SCH$_2$Ph), 4.00 (s, 2H, COCH$_2$), 2.27 (s, 3H, CH$_3$); $^{13}$C NMR (101 MHz, DMSO-$d_6$) δ 193.4, 168.4, 159.1, 158.8, 144.7, 137.3, 134.0, 129.9, 129.6, 129.4, 128.3, 127.7, 74.1, 59.3, 53.2, 37.8, 21.2; HRMS (ESI) $m/z$ calcd for C$_{21}$H$_{22}$N$_5$OS$_4$ ([M+H]$^+$): 488.0702, found: 488.0696.

### N-(5-((4-chlorobenzyl)thio)-1,3,4-thiadiazol-2-yl)-2-(5-phenyl-6-thioxo-1,3,5-thiadiazinan-3-yl)acetamide (5d)

White solid, m.p.179–181 °C, yield 69%; $^1$H NMR (400 MHz, DMSO-$d_6$) δ 7.49–7.32 (m, 7H, PhH), 7.24 (d, $J$ = 7.5 Hz, 2H, PhH), 4.75 (s, 2H, NCH$_2$N), 4.68 (s, 2H, SCH$_2$N), 4.49 (d, $J$ = 11.2 Hz, 2H, SCH$_2$Ph), 4.00 (s, 2H, COCH$_2$); $^{13}$C NMR (101 MHz, DMSO-$d_6$) δ 193.4, 168.5, 159.3, 158.3, 144.7, 136.5, 132.6, 131.3, 129.9, 129.0, 128.3, 127.7, 74.1, 59.3, 53.2, 37.1; HRMS (ESI) $m/z$ calcd for C$_{20}$H$_{18}$ClN$_5$NaOS$_4$([M+Na]$^+$): 529.9975, found: 529.9970.

### 2-(5-benzyl-6-thioxo-1,3,5-thiadiazinan-3-yl)-N-(5-(ethylthio)-1,3,4-thiadiazol-2-yl)acetamide (6a)

White solid, m.p.179–181 °C, yield 79%; $^1$H NMR (400 MHz, DMSO-$d_6$) δ 12.50 (s, 1H, CONH), 7.35 (d, $J$ = 7.5 Hz, 2H, PhH), 7.27 (t, $J$ = 7.6 Hz, 2H, PhH), 7.12 (t, $J$ = 7.3 Hz, 1H, PhH), 5.28 (s, 2H, NCH$_2$Ph), 4.58 (s, 2H, NCH$_2$N), 4.50 (s, 2H, SCH$_2$N), 3.69 (s, 2H, COCH$_2$), 3.22 (q, $J$ = 7.3 Hz, 2H, CH$_2$CH$_3$), 1.34 (t, $J$ = 7.3 Hz, 3H, CH$_3$); $^{13}$C NMR

(101 MHz, DMSO-$d_6$) δ 192.2, 167.8, 159.1, 158.6, 136.0, 129.0, 128.4, 127.9, 68.9, 59.3, 53.3, 52.7, 28.6, 15.2; HRMS (ESI) *m/z* calcd for $C_{16}H_{20}N_5OS_4$ ([M+H]$^+$): 426.0545, found: 426.0542.

### 2-(5-benzyl-6-thioxo-1,3,5-thiadiazinan-3-yl)-*N*-(5-(benzylthio)-1,3,4-thiadiazol-2-yl)acetamide (6b)

White solid, m.p.160–162 °C, yield 79%; $^1$H NMR (400 MHz, DMSO-$d_6$) δ 12.46 (s, 1H, CONH), 7.41 (d, *J* = 7.4 Hz, 2H, PhH), 7.33 (s, 4H, PhH), 7.30–7.21 (m, 3H, PhH), 7.08 (t, *J* = 7.3 Hz, 1H, PhH), 5.27 (s, 2H, NCH$_2$Ph), 4.57 (s, 2H, SCH$_2$Ph), 4.48 (s, 4H, NCH$_2$NCH$_2$S), 3.66 (s, 2H, COCH$_2$); $^{13}$C NMR (101 MHz, DMSO-$d_6$) δ 192.2, 167.9, 159.0, 158.4, 137.2, 136.0, 129.4, 129.0, 128.4, 128.1, 127.9, 68.89, 59.3, 53.4, 52.7, 38.0; HRMS (ESI) *m/z* calcd for $C_{21}H_{22}N_5OS_4$ ([M+H]$^+$): 488.0702, found: 488.0695.

### 2-(5-benzyl-6-thioxo-1,3,5-thiadiazinan-3-yl)-*N*-(5-((4-methylbenzyl)thio)-1,3,4-thiadiazol-2-yl)acetamide (6c)

White solid, m.p.179–181 °C, yield 71%; $^1$H NMR (400 MHz, DMSO-$d_6$) δ 12.43 (s, 1H, CONH), 7.34 (d, *J* = 7.4 Hz, 2H, PhH), 7.31–7.21 (m, 4H, PhH), 7.14 (d, *J* = 7.2 Hz, 2H, PhH), 7.09 (t, *J* = 7.0 Hz, 1H, PhH), 5.28 (s, 2H, NCH$_2$Ph), 4.57 (s, 2H, SCH$_2$Ph), 4.48 (s, 2H, NCH$_2$N), 4.44 (s, 2H, SCH$_2$N), 3.66 (s, 2H, COCH$_2$), 2.27 (s, 3H, CH$_3$); $^{13}$C NMR (101 MHz, DMSO-$d_6$) δ 192.2, 167.9, 156.0, 158.6, 137.3, 136.0, 134.1, 129.6, 129.4, 129.0, 128.5, 127.9, 68.9, 59.3, 53.4, 52.7, 37.9, 21.2; HRMS (ESI) *m/z* calcd for $C_{22}H_{24}N_5OS_4$ ([M+H]$^+$): 502.0858, found: 502.0850.

### 2-(5-benzyl-6-thioxo-1,3,5-thiadiazinan-3-yl)-*N*-(5-((4-chlorobenzyl)thio)-1,3,4-thiadiazol-2-yl)acetamide (6d)

White solid, m.p.175–177 °C, yield 72%; $^1$H NMR (400 MHz, DMSO-$d_6$) δ 7.41 (dd, *J* = 16.2, 7.8 Hz, 4H, PhH), 7.34 (d, *J* = 7.4 Hz, 2H, PhH), 7.24 (t, *J* = 7.3 Hz, 2H, PhH), 7.07 (t, *J* = 7.3 Hz, 1H, PhH), 5.28 (s, 2H, NCH$_2$Ph), 4.57 (s, 2H, SCH$_2$Ph), 4.48 (s, 4H, NCH$_2$NCH$_2$S), 3.66 (s, 2H, COCH$_2$); $^{13}$C NMR (101 MHz, DMSO-$d_6$) δ 192.12, 167.9, 159.2, 158.0, 136.5, 136.0, 132.6, 131.33, 129.0, 129.0, 128.4, 127.9, 68.9, 59.3, 53.4, 52.7, 37.2; HRMS (ESI) *m/z* calcd for $C_{21}H_{20}ClN_5NaOS_4$ ([M+Na]$^+$): 544.0132, found: 544.0125.

### *N*-(5-(ethylthio)-1,3,4-thiadiazol-2-yl)-2-(5-(4-fluorophenyl)-6-thioxo-1,3,5-thiadiazinan-3-yl)acetamide (7a)

White solid, m.p.173–175 °C, yield 41%; $^1$H NMR (400 MHz, DMSO-$d_6$) δ 7.30 (d, *J* = 6.9 Hz, 4H, PhH), 4.76 (s, 2H, NCH$_2$), 4.69 (s, 2H, SCH$_2$N), 4.01 (s, 2H, COCH$_2$), 3.22 (q, *J* = 7.3 Hz, 2H, CH$_2$CH$_3$), 1.33 (t, *J* = 7.3 Hz, 3H, CH$_3$); $^{13}$C NMR (101 MHz, DMSO-$d_6$) δ 194.0, 168.4, 159.3, 158.7, 140.8, 130.0, 129.9, 116.9, 116.6, 74.1, 59.3, 53.2, 28.5, 15.2; HRMS (ESI) *m/z* calcd for $C_{16}H_{19}FN_5OS_4$ ([M+H]$^+$): 430.0295, found: 430.0291.

### N-(5-(benzylthio)-1,3,4-thiadiazol-2-yl)-2-(5-(4-fluorophenyl)-6-thioxo-1,3,5-thiadiazinan-3-yl)acetamide (7b)

White solid, m.p.163–165 °C, yield 57%; $^1$H NMR (400 MHz, DMSO-$d_6$) δ 12.73 (s, 1H, CONH), 7.44 (d, $J$ = 7.0 Hz, 2H, PhH), 7.40–7.30 (m, 7H, PhH), 4.79 (s, 2H, NCH$_2$N), 4.72 (s, 2H, SCH$_2$N), 4.53 (s, 2H, SCH$_2$Ph), 4.05 (s, 2H, COCH$_2$); $^{13}$C NMR (101 MHz, DMSO-$d_6$) δ 194.0, 168.4, 159.1, 158.8, 140.8, 137.2, 130.0, 129.9, 129. 5, 129.0, 128.1, 116.9, 116.6, 74.1, 59.4, 53.2, 38.0; HRMS (ESI) $m/z$ calcd for C$_{20}$H$_{19}$FN$_5$OS$_4$ ([M+H]$^+$): 492.0451, found: 492.0444.

### 2-(5-(4-fluorophenyl)-6-thioxo-1,3,5-thiadiazinan-3-yl)-N-(5-((4-methyl-benzyl)thio)-1,3,4-thiadiazol-2-yl)acetamide (7c)

White solid, m.p.170–171 °C, yield 49%; $^1$H NMR (400 MHz, DMSO-$d_6$) δ 7.29 (t, $J$ = 7.8 Hz, 6H, PhH), 7.13 (d, $J$ = 7.6 Hz, 2H, PhH), 4.75 (s, 2H, NCH$_2$N), 4.68 (s, 2H, SCH$_2$N), 4.44 (s, 2H, SCH$_2$Ph), 4.00 (s, 2H, COCH$_2$), 2.27 (s, 3H, CH$_3$); $^{13}$C NMR (101 MHz, DMSO-$d_6$) δ 194.0, 168.5, 159.1, 158.8, 140.8, 137.3, 134.0, 130.0, 129.9, 129.6, 129.4, 116.9, 116.6, 74.1, 59.4, 53.3, 37.8, 21.2; HRMS (ESI) $m/z$ calcd for C$_{21}$H$_{21}$FN$_5$OS$_4$ ([M+H]$^+$): 506.0608, found: 506.0598.

### N-(5-((4-chlorobenzyl)thio)-1,3,4-thiadiazol-2-yl)-2-(5-(4-fluorophenyl)-6-thioxo-1,3,5-thiadiazinan-3-yl)acetamide (7d)

White solid, m.p.174–175 °C, yield 48%; $^1$H NMR (400 MHz, DMSO-$d_6$) δ 7.40 (dd, $J$ = 16.1, 7.9 Hz, 4H, PhH), 7.30 (d, $J$ = 6.5 Hz, 4H, PhH), 4.75 (s, 2H, NCH$_2$N), 4.68 (s, 2H, SCH$_2$N), 4.48 (s, 2H, SCH$_2$Ph), 4.00 (s, 2H, COCH$_2$); $^{13}$C NMR (101 MHz, DMSO-$d_6$) δ 194.0, 168.5, 159.3, 158.3, 140.8, 136.5, 132.6, 131.3, 130.0, 129.9, 129.0, 116.9, 116.6, 74.1, 59.4, 53.3, 37.1; HRMS (ESI) $m/z$ calcd for C$_{20}$H$_{18}$ClFN$_5$OS$_4$ ([M+H]$^+$): 526.0061, found: 526.0049.

### N-(5-(ethylthio)-1,3,4-thiadiazol-2-yl)-2-(5-methyl-6-thioxo-1,3,5-thiadiazinan-3-yl)acetamide (8a)

White solid, m.p.159–161 °C, yield 45%; $^1$H NMR (400MHz, DMSO-$d_6$) δ 4.55 (s, 4H, NCH$_2$NCH$_2$S), 3.83 (s, 2H, COCH$_2$), 3.37 (s, 3H, NCH$_3$), 3.23 (q, $J$ = 7.3 Hz, 2H, CH$_2$CH$_3$), 1.35 (t, $J$ = 7.3 Hz, 3H, CH$_2$CH$_3$); $^{13}$C NMR (101 MHz, DMSO-$d_6$) δ 190.5, 168.5, 159.2, 158.8, 71.8, 59.3, 58.8, 53.3, 28.6, 15.2; HRMS (ESI) $m/z$ calcd for C$_{10}$H$_{16}$N$_5$OS$_4$ ([M+H]$^+$): 350.0232, found: 350.0230.

### N-(5-(benzylthio)-1,3,4-thiadiazol-2-yl)-2-(5-methyl-6-thioxo-1,3,5-thiadiazinan-3-yl)acetamide (8b)

White solid, m.p.166–168 °C, yield 66%; $^1$H NMR (400MHz, DMSO-$d_6$) δ 12.45 (s, 1H, CONH), 7.41 (d, $J$ = 7.4 Hz, 2H, PhH), 7.34 (t, $J$ = 7.3 Hz, 2H, PhH), 7.31–7.24 (m, 1H, PhH), 4.54 (s, 4H, NCH$_2$NCH$_2$S), 4.49 (s, 2H, SCH$_2$Ph), 3.82 (s, 2H, COCH$_2$), 3.37 (s, 3H, NCH$_3$); $^{13}$C NMR (101 MHz, DMSO-$d_6$) δ 190.5, 168.6, 159.2, 158.6, 137.2, 129.5, 129.0, 128.1, 71.8, 58.8, 53.3, 38.0; HRMS (ESI) $m/z$ calcd for C$_{15}$H$_{17}$N$_5$NaOS$_4$ ([M+Na]$^+$): 434.0208, found: 434.0204.

### 2-(5-methyl-6-thioxo-1,3,5-thiadiazinan-3-yl)-*N*-(5-((4-methylbenzyl)thio)-1,3,4-thiadiazol-2-yl)acetamide (8c)

White solid, m.p.168–170 °C, yield 56%; $^{1}$H NMR (400MHz, DMSO-$d_6$) δ 7.28 (d, $J$ = 7.5 Hz, 2H, PhH), 7.13 (d, $J$ = 7.7 Hz, 2H, PhH), 4.53 (s, 4H, NCH$_2$NCH$_2$S), 4.44 (s, 2H, SCH$_2$Ph), 3.81 (s, 2H, COCH$_2$), 3.36 (s, 3H, NCH$_3$), 2.27 (s, 3H, PhCH$_3$); $^{13}$C NMR (101 MHz, DMSO-$d_6$) δ 190.5, 168.6, 159.2, 158.8, 137.3, 134.0, 129.6, 129.4, 71.8, 58.8, 53.3, 37.8, 21.2; HRMS (ESI) $m/z$ calcd for C$_{16}$H$_{20}$N$_5$OS$_4$([M+H]$^{+}$): 426.0545, found: 426.0541.

### *N*-(5-((4-chlorobenzyl)thio)-1,3,4-thiadiazol-2-yl)-2-(5-methyl-6-thioxo-1,3,5-thiadiazinan-3-yl)acetamide (8d)

White solid, m.p.161–163 °C, yield 56%; $^{1}$H NMR (400 MHz, DMSO-$d_6$) δ 12.49 (s, 1H, CONH), 7.41 (q, $J$ = 8.1 Hz, 4H, PhH), 4.53 (s, 4H, NCH$_2$NCH$_2$S), 4.48 (s, 2H, SCH$_2$Ph), 3.81 (s, 2H, COCH$_2$), 3.36 (s, 3H, CH$_3$); $^{13}$C NMR (101 MHz, DMSO-$d_6$) δ 190.5, 168.7, 159.5, 158.2, 136.5, 132.6, 131.3, 129.0, 71.8, 58.8, 53.4, 37.1; HRMS (ESI) $m/z$ calcd for C$_{15}$H$_{16}$ClN$_5$NaOS$_4$ ([M+Na]$^{+}$): 467.9819, found: 467.9814.

### Crystal structure determination

The title compound **8d** was recrystallized from a mixture of DMF and methanol (V:V = 1:1) to obtain a suitable single crystal. The X-ray single crystal diffraction data was collected on an Agilent Super Nova (Dual, Cu at zero, AtlasS2) single crystal diffractometer at 100.00 (10) K with the monochromatized MoKα radiation (λ = 0.71073 Å) using $w$ scan mode. The CrysAlisPro program was used to integrate the diffraction profile. The structure was solved directly and optimized by using full matrix least square method via SHELXL (*Sheldrick, 1997*). All the non-hydrogen atoms were refined by full-matrix least-squares technique on $F^2$ with anisotropic thermal parameters. All the hydrogen atoms were positioned geometrically and refined using a riding model. PLATON program was used for the structural analysis and the diamond program was used for the drawings (*Spek, 2003*).

### Antibacterial activities test in vitro

The antibacterial activities of title compounds against *X. oryzae pv. oryzae* and *X. oryzae pv. oryzicola* were evaluated by the turbidimeter test (*Li et al., 2014*; *Wang et al., 2013*; *Xu et al., 2012*). The compounds were dissolved in dimethylsulfoxide (DMSO) and diluted with water (containing 0.1% Tween-20) to obtain a solution with a final concentration of 100 and 50 μg/mL by adding different amounts water. DMSO in sterile distilled water served as a blank control, thiodiazole-copper served as positive control. Approximately one mL of sample liquid was added to the nontoxic nutrient broth (NB, 3.0 g of beef extract, 5.0 g of peptone, 1.0 g of yeast powder, 10.0 g of glucose, and 1,000 mL of distilled water, pH 7.0–7.2) liquid medium in four mL tubes. Then, about 40 μL of solvent NB containing *X. oryzae pv. oryzae* or *X. oryzae pv. oryzicola* was added to five mL of solvent NB containing the test compounds and positive control. The inoculated test tubes were incubated at 28 ± 1 °C and continuously cultured shakily at 180 rpm for 2–3 days.

The growth of the cultures was monitored on a microplate reader by measuring the optical density at 600 nm ($OD_{600}$) given by turbidity$_{corrected\ values}$ = $OD_{bacterium}$ − $OD_{no\ bacterium}$, and then the inhibition rate $I$ (%) was calculated by $I = (C_{tur} − T_{tur})/C_{tur} \times 100\%$. $C_{tur}$ is the corrected turbidity values of bacterial growth on untreated NB (blank control), and $T_{tur}$ is the corrected turbidity values of bacterial growth on treated NB.

## Antifungal activities test in vitro

*Rhizoctonia solani* and *Fusarium graminearum*, which were the representative plant pathogenic fungi, were chosen as the test strains. The antifungal activities of title compounds against *R. solani* and *F. graminearum* in vitro were tested by the mycelium growth rate method (*Wang et al., 2017*; *Chen et al., 2012*). Different doses of compounds were dissolved in DMSO and mixed with sterile molten potato dextrose agar (PDA, 200 g potato, 20 g glucose, 18 g agar, add water and boil to prepare 1,000 mL solution) medium to obtain a final concentration of 100 and 50 μg/mL. DMSO in sterile distilled water was used as the negative control, commercial fungicide hymexazol was selected as a positive control. Place fungi mycelia disks (four mm diameter) at the center of Petri dishes in a sterile environment and the treatments were incubated in the dark at 25 ± 1 °C. Each treatment was produced in three replicates. The diameters of the sample colonies were measured, when the colonies in the control experiment covered two-thirds of the culture dishes. Inhibitory percentages of the title compounds in vitro on these fungi were calculated as $I = [(C − T)/(C − 4)] \times 100\%$, where $I$ was the inhibition rate (%), $C$ was the diameter (mm) of the fungal colony on the negative control group, and $T$ was the diameter (mm) of the fungal colony on the experimental group.

## RESULTS

### Spectral characteristic of title compounds

The structures of the title compounds were confirmed by $^{1}$H NMR, $^{13}$C NMR and HRMS data. Here, we take the compound **5a** as the example to analyze the molecular structure. In the $^{1}$H NMR spectra of the compound **5a** (Fig. S1), the broad singlet proton peak was at 12.73 ppm of the amide group and (–NCH$_2$N–, –SCH$_2$N–) proton of the 1,3,5-thiadiazinethione were about 4.76 and 4.69 ppm, respectively. Other (–CH$_2$C=O) protons were at around 4.02 ppm. The $^{13}$C NMR peak of the thiophenone group (–C=S) emerged between 194.03 and 190.48 ppm, and carbonyl group (–C=O) peaks were in the range of 168.65–167.83 ppm (Fig. S2). In addition, the carbon atom peaks were at two or five position of the 1,3,4-thiadiazole at 159.47–158.02 ppm. The HRMS spectra (Fig. S3) exhibited that the compound **5a** was peaking at 412.0388 ([M+H]$^{+}$). For more characteristic information about the compounds **5–8**, please refer to the Supplemental Information.

### X-ray crystal structure of compound 8d

The structure of compound **8d** was further confirmed using the single crystal X-ray analysis. The corresponding crystal structure and crystal packing diagrams were shown in Figs. 4 and 5, respectively. The hydrogen bonds were given in Table 1. As shown in Fig. 4,

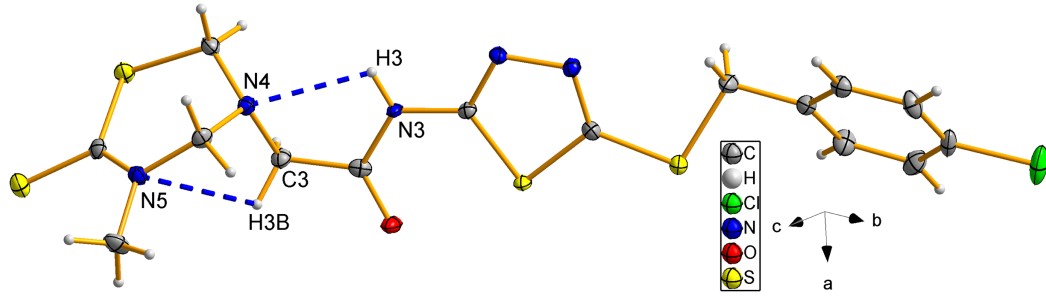

**Figure 4** The molecular ellipsoid of compound **8d**.

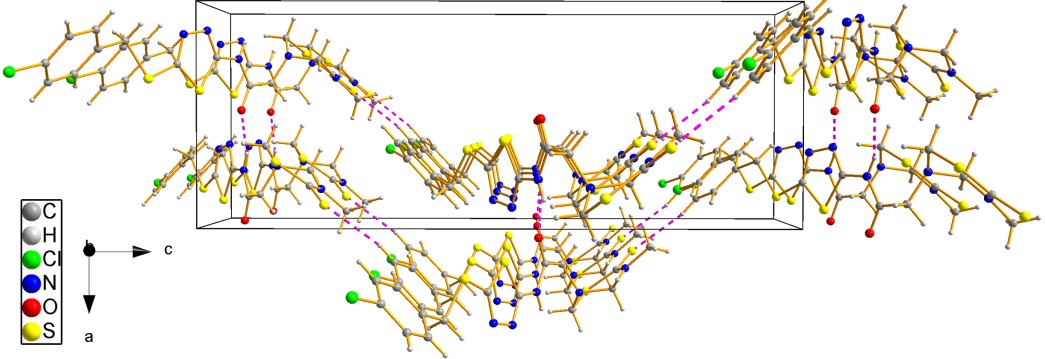

**Figure 5** Crystal packing diagram of compound **8d**.

**Table 1** Hydrogen bond distances (Å) and angles (°) of compound 8d.

| D–H⋯A | d(D–H) | d(H⋯A) | d(D⋯A) | ∠(DHA) |
|---|---|---|---|---|
| N(3)–H(3)⋯N(4) | 0.84(4) | 2.41(4) | 2.759(4) | 106(3) |
| C(3)–H(3B)⋯N(5) | 0.9700 | 2.6100 | 3.008(5) | 105.00 |
| N(3)–H(3)⋯O(1)[a] | 0.84(4) | 2.00(4) | 2.809(4) | 160(4) |
| C(12)–H(12)⋯S(3)[b] | 0.9300 | 2.8600 | 3.769(4) | 165.00 |

**Notes:**
[a] Symmetry code: −1/2 + x, 1/2 − y, z.
[b] Symmetry code: 2 − x, 1 − y, −1/2 + z.

the intramolecular hydrogen bond N(3)–H(3)⋯N(4) formed a new five-membered ring with two other C atoms. In addition, the intramolecular hydrogen bond C(3)–H(3B)⋯N(5), together with thiadiazinthion ring constituted a new bridge ring. In the packing diagram of the compound **8d** (Fig. 5), the molecules connected each other through intermolecular hydrogen bonds N(3)–H(3)⋯O(1) and C(12)–H(12)⋯S(3) (Table 1). Among them, intermolecular hydrogen bond C(12)–H(12)⋯S(3) connected different molecules to form the molecular chains, while N(3)–H(3)⋯O(1) connected different chains to form planes, eventually the spatial network was formed. Crystallographic data were deposited with the Cambridge Crystallographic Data Centre: deposition number CCDC 1912576.

**Table 2 Inhibition rates of title compounds against phytopathogenic microorganisms[a].**

| Compd | R[1] | R[2] | Xanthomonas oryzae pv. oryzicola | | Xanthomonas oryzae pv. oryzae | | Rhizoctonia solani | | Fusarium graminearum | |
|---|---|---|---|---|---|---|---|---|---|---|
| | | | 100 µg/mL | 50 µg/mL | 100 µg/mL | 50 µg/mL | 100 µg/mL | 50 µg/mL | 100 µg/mL | 50 µg/mL |
| 5a | Ph | Me | 17 ± 0.47 | 14 ± 3.08 | 30 ± 2.02 | 17 ± 1.66 | 35 ± 3.68 | 10 ± 2.24 | 34 ± 4.10 | 18 ± 3.05 |
| 5b | Ph | Ph | 22 ± 1.41 | 18 ± 4.13 | 24 ± 4.10 | 15 ± 3.48 | 14 ± 1.76 | 6 ± 4.65 | 15 ± 3.35 | 7 ± 3.55 |
| 5c | Ph | 4-MePh | 25 ± 3.54 | 20 ± 1.71 | 22 ± 0.11 | 14 ± 2.55 | 21 ± 1.94 | 5 ± 1.72 | 17 ± 3.69 | 6 ± 1.76 |
| 5d | Ph | 4-ClPh | 13 ± 2.15 | 10 ± 2.02 | 24 ± 0.34 | 10 ± 0.10 | 27 ± 2.69 | 15 ± 3.40 | 20 ± 4.18 | 7 ± 2.29 |
| 6a | Bn | Me | 17 ± 4.18 | 10 ± 1.15 | 38 ± 1.06 | 26 ± 0.55 | 23 ± 1.69 | 13 ± 2.74 | 23 ± 1.39 | 12 ± 2.96 |
| 6b | Bn | Ph | 14 ± 3.31 | 11 ± 2.11 | 29 ± 4.18 | 19 ± 0.69 | 18 ± 1.47 | 14 ± 3.27 | 26 ± 4.44 | 18 ± 2.61 |
| 6c | Bn | 4-MePh | 17 ± 0.92 | 11 ± 1.52 | 20 ± 4.11 | 13 ± 0.43 | 29 ± 2.44 | 22 ± 4.10 | 26 ± 4.71 | 12 ± 1.88 |
| 6d | Bn | 4-ClPh | 9 ± 4.74 | 5 ± 1.13 | 25 ± 0.43 | 16 ± 0.55 | 30 ± 2.20 | 15 ± 1.83 | 32 ± 3.41 | 15 ± 2.79 |
| 7a | 4-FPh | Me | 14 ± 0.01 | 12 ± 1.35 | 35 ± 0.91 | 25 ± 0.02 | 21 ± 1.69 | 10 ± 2.24 | 15 ± 3.66 | 11 ± 2.69 |
| 7b | 4-FPh | Ph | 13 ± 3.42 | 6 ± 3.77 | 26 ± 2.97 | 20 ± 0.70 | 22 ± 1.67 | 9 ± 1.70 | 19 ± 3.51 | 6 ± 2.43 |
| 7c | 4-FPh | 4-MePh | 15 ± 3.85 | 9 ± 2.10 | 24 ± 0.36 | 17 ± 4.63 | 8 ± 1.67 | 0 | 14 ± 3.47 | 6 ± 3.03 |
| 7d | 4-FPh | 4-ClPh | 10 ± 7.54 | 5 ± 4.44 | 25 ± 0.91 | 13 ± 2.30 | 46 ± 2.44 | 28 ± 4.10 | 14 ± 1.75 | 7 ± 2.04 |
| 8a | Me | Me | 30 ± 2.58 | 25 ± 5.87 | 56 ± 2.02 | 29 ± 2.03 | 100 | 62 ± 6.37 | 67 ± 7.14 | 43 ± 5.05 |
| 8b | Me | Ph | 16 ± 4.13 | 12 ± 3.61 | 32 ± 0.36 | 26 ± 4.63 | 50 ± 5.89 | 36 ± 3.00 | 30 ± 5.80 | 16 ± 3.39 |
| 8c | Me | 4-MePh | 18 ± 4.68 | 15 ± 4.40 | 20 ± 5.12 | 12 ± 3.49 | 19 ± 4.76 | 13 ± 2.26 | 20 ± 1.47 | 9 ± 2.59 |
| 8d | Me | 4-ClPh | 13 ± 2.80 | 10 ± 4.00 | 25 ± 4.12 | 19 ± 4.05 | 22 ± 6.15 | 7 ± 2.59 | 21 ± 1.52 | 12 ± 3.49 |
| TC[b] | – | – | 18 ± 4.77 | 10 ± 2.79 | 40 ± 1.02 | 29 ± 4.43 | – | – | – | – |
| HY[b] | – | – | – | – | – | – | 47 ± 6.94 | 37 ± 7.70 | 66 ± 4.09 | 43 ± 1.89 |

Notes:
[a] Average of three replicates.
[b] A commercial agricultural bacterial thiodiazole-copper and hymexazol were used for comparison of antibacterial activities.

## Antibacterial bioassays of title compounds in vitro

All the target compounds were tested for the in vitro antibacterial activity against *X. oryzae pv. oryzicola* and *X. oryzae pv. oryzae* to investigate the biological activity. The preliminary bioassay results demonstrated that all compounds had certain antibacterial activity against *X. oryzae pv. oryzicola* and *X. oryzae pv. oryzae* at 100 and 50 µg/mL (Table 2). For example, the inhibitory rates of compounds **5b**, **5c**, **8c** and **8a** against *X. oryzae pv. oryzicola*, respectively, were 22%, 22%, 18% and 30% at 100 µg/mL, which are better than that of thiodiazole-copper (18%). Otherwise, compounds **5a–5d**, **6a–6c**, **7a** and **8a–8c** exhibited certain inhibitory activities against *X. oryzae pv. oryzicola* comparing with the thiodiazole-copper at 50 µg/mL. In addition, the title compounds **5a**, **6a**, **7a**, **8a** and **8b** also showed certain activities against *X. oryzae pv. oryzae* at 50 and 100 µg/mL. Among them, **8a** exhibited better antibacterial activities than thiodiazole-copper. As can be seen, compound **8a** was the best inhibitor among all the compounds, not only has antibacterial activity against *X. oryzae pv. oryzae*, but also has a certain inhibitory effect against *X. oryzae pv. oryzicola*.

## Antifungal bioassays of title compounds in vitro

The in vitro inhibitory activity against *R. solani* and *F. graminearum* were tested. Table 2 shows that compounds **5a**, **6d**, **7d** and **8b** possessed certain activities against both *R. solani*

**Table 3** EC$_{50}$ values of the title compound 8a against *Rhizoctonia solani* and *Fusarium graminearum*.

| Compd | Strains | Regression equation | r | EC$_{50}$ (µg/mL)[a] |
|---|---|---|---|---|
| **8a** | *Rhizoctonia solani* | $y = 2.1218x + 1.7578$ | 0.9925 | 33.70 ± 0.24 |
| **8a** | *Fusarium graminearum* | $y = 1.2648x + 2.5363$ | 0.9654 | 88.70 ± 0.49 |
| Hymexazol[b] | *Rhizoctonia solani* | $y = 1.5892x + 2.0968$ | 0.9869 | 67.10 ± 0.24 |
| Hymexazol | *Fusarium graminearum* | $y = 1.8967x + 1.6815$ | 0.9986 | 56.19 ± 1.68 |

**Notes:**
[a] Average of three replicates.
[b] Hymexazol, the agricultural fungicide, was used for the comparison of antifungal effects.

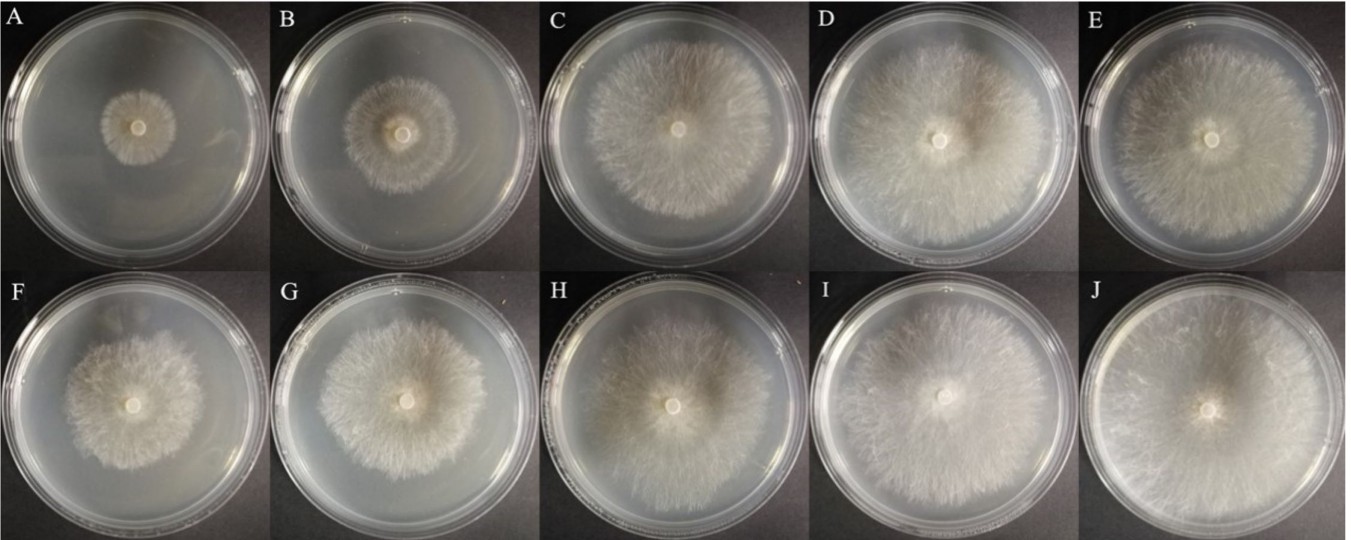

**Figure 6** Anti-*Rhizoctonia solani* effects of the bioactive compounds 8a and hymexazol. (A) **8a** at 50 µg/mL, (B) **8a** at 25 µg/mL, (C) **8a** at 12.5 µg/mL, (D) **8a** at 6.25 µg/mL, (E) **8a** at 3.125 µg/mL, (F) hymexazol at 50 µg/mL, (G) hymexazol at 25 µg/mL, (H) hymexazol at 12.5 µg/mL, (I) hymexazol at 6.25 µg/mL and (J) hymexazol at 3.125 µg/mL.

and *F. graminearum*. At the concentration of 100 µg/mL, compound **7d** and **8b** showed comparative activity (46%, 50%) against *R. solani*, which was comparable with the commercial drug hymexazol (47%). Amongst all the compounds, **8a** displayed the best inhibitory activity (100% at 100 µg/mL and 62% at 50 µg/mL) against *R. solani*, which was even better than that of hymexazol (47% at 100 µg/mL and 37% at 50 µg/mL) (Table 2). Moreover, compound **8a** also exhibited good activity (67% at 100 µg/mL and 43% at 50 µg/mL) against *F. graminearum*, which was approximate with the activity (66% at 100 µg/mL and 43% at 50 µg/mL) of hymexazol (Table 2).

Afterward, the EC$_{50}$ values of compound **8a** and hymexazol against *R. solani* and *F. graminearum* were tested, respectively, which were shown in Table 3 and Fig. 6. The EC$_{50}$ value of **8a** against *F. graminearum* was 88.7 µg/mL, which was higher than that of hymexazol (56.19 µg/mL). But the compound **8a** showed remarkable activity against *R. solani* with an EC$_{50}$ value of 33.70 µg/mL, which was superior to that of hymexazol (67.10 µg/mL).

## DISCUSSION

### Synthesis

The core intermediates **3** were obtained by three steps according to the reported method (*Echemendía et al., 2017*) with small modifications. The 5-amino-1,3,4-thiadiazole-2-thiol reacted with substituted ethyl bromide or substituted benzyl chloride in acetonitrile to form the intermediates **4** with the participation of potassium carbonate. To our disappointment, the reaction time was long and the yield was low (about 40%). When DMF was used as the solvent, the reaction was completed in 2–8 h at the same temperature, and the yield was increased to 80%. Optimal conditions for the reaction of intermediates **3** and **4** to produce **5–8**. First, an attempt was made to convert intermediates **3** to the corresponding acid chlorides and then reacted with **4**. Unfortunately, the reaction failed. Subsequently, different methods, such as the use of 1-(3-dimethylaminopropyl)-3-ethylcarbodiimide (EDCI) hydrochloride/1-hydroxybenzotriazole/triethylamine ($Et_3N$), EDCI/4-dimethylaminopyridine/$Et_3N$, gave no reaction. However, it was good that the TBTU/$Et_3N$ could complete the reaction in 2 h at room temperature. The result showed that the reaction was carried out well in the condition of TBTU/$Et_3N$, which had many advantages, such as the short reaction time, the reaction mild conditions and the simple and convenient post-treatment. In addition, the yield was 41–79%. It is observed that when $R^1$ was 4-FPh, the yield is lower than that of Phenyl, which might be related to the electron-withdrawing group on the benzene ring.

### Antimicrobial activity

From Table 2, it can be concluded that the changes of the substituents $R^1$ and $R^2$ have a certain effect on the biological activity of the compounds.

For *X. oryzae pv. oryzicola*, it was seen that compounds **5**, **8** displayed slightly higher activity than compounds **6** and **7** series. This indicated that when the $R^1$ substituent group was substituted with a methyl or phenyl, the corresponding compounds generally exhibited better anti-*X. oryzae pv. oryzicola* effects than those compounds bearing a benzyl or 4-fluorophenyl. On the whole, **5a**, **5b**, **5c**, **6a**, **6c**, **8a** and **8c** had better activity than the others. Among them, the inhibition rates of **5a**, **6a**, **6c** and **8c** (17%, 17%, 17% and 18%) were approximate with the inhibition rate of commercial bactericide thiodiazole-copper (18%) at 100 μg/mL. This result also applied to the concentration of 50 μg/mL. At 100 μg/mL, the activity of compounds **5b** and **5c** (22% and 25%) was slightly higher than that of thiodiazole-copper. This difference was even great at 50 μg/mL, the inhibition rates of **5b**, **5c** and thiodiazole-copper were 18%, 20% and 10%, respectively. The activity of **8a** (30%, 100 μg/mL; 25%, 50 μg/mL) was significantly better than that of thiodiazole-copper.

Bacteria *X. oryzae pv. oryzae* was more sensitive than *X. oryzae pv. oryzicola* for all the test compounds, including thiodiazole-copper, which appears in Table 2. For *X. oryzae pv. oryzae*, **5a** was the most active compound in compounds **5**, for which the inhibition rate of **5a** (30%) was significantly higher than the others (22–24%). A similar pattern occurs in compounds **6**, **7** and **8**, that was **6a**, **7a** and **8a** were the most active compounds in

their respective series. It could be concluded that the substitute $R^2$ played a role in the activity. When the $R^1$ was the same substitute, title compounds bearing a methyl at $R^2$ position exhibited more obvious anti-*X. oryzae pv. oryzae* effects than the others, which contained benzyl, 4-methyl phenyl or 4-chlorphenyl moiety at $R^2$ position. Among all the target compounds, **6a**, **7a** and **8a** showed higher inhibitory activity than the others, both at 100 and 50 µg/mL. Compound **8a** was the best inhibitor and showed the inhibition rates of 56% and 29% at the concentration of 100 and 50 µg/mL, respectively, which was compared with the activity of 40% (at 100 µg/mL) and 29% (at 100 µg/mL) of thiodiazole-copper.

In general, most of the compounds showed a higher antifungal activity toward *R. solani* than the *F. graminearum*. It can be found that the changes in substituent groups of title compounds greatly influenced their antifungal effects. The structure–activity relationships showed three general rules. First, overall, the inhibitory activities of the target compounds against *R. solani* were higher than that of *F. graminearum*, except for compounds **6b** and **7c**. Second, when $R^1$ was Me, that was to say **8a–8d**, exhibited better antifungal effects than those compounds when $R^1$ was Ph (**5a–5d**), 4-FPh (**7a–7d**) or Bn (**6a–6b**). Third, when $R^2$ was 4-Clphenyl, the activities of the corresponding compounds were superior to these compounds with $R^2$ was 4-Mephenyl. For example, the compounds fell into order by inhibitory rate as **5d** > **5c**, **6d** > **6c**, **7d** > **7c** and **8d** > **8c**, against both fungi at the concentration of 100 µg/mL.

From the antimicrobial activity against *X. oryzae pv. oryzicola*, *X. oryzae pv. oryzae*, *R. solani* and *F. graminearum* (Tables 2 and 3), it was indicated that the changes of substituent groups $R^1$ and $R^2$ greatly influenced the antimicrobial activities of the title compounds. In this paper, it was found that when $R^1$ and $R^2$ were the methyl group, the target compound (**8a**) showed distinct antibacterial and antifungal activities. The compound **8a** showed better antimicrobial activities against *X. oryzae pv. oryzicola* and *R. solani* than the commercial thiodiazole-copper and hymexazol, respectively. The activity results provided a direction for further molecular structure optimization of 1,3,5-thiadiazine-2-thione derivatives. The related research is continuing in our laboratory.

## CONCLUSIONS

Sixteen novel 1,3,5-thiadiazine-2-thione derivatives containing a 1,3,4-thiadiazole group was designed, synthesized, characterized and screened for the antibacterial effects against *X. oryzae pv. oryzicola* and *X. oryzae pv. oryzae* as well as the antifungal effects against *R. solani* and *F. graminearum*. The antimicrobial bioassays showed that some title compounds displayed valuable antibacterial and antifungal activities. The compound **8a**, in which $R^1$ and $R^2$ were the methyl group, was one of the most prominent activities against all the test microbials. The compound **8a** possessed meaningful antibacterial effects against *X. oryzae pv. oryzicola*, with inhibition rates of 30% at 100 µg/mL and 25% at 50 µg/mL, respectively, which were higher than that of thiodiazole-copper (18% at 100 µg/mL and 10% at 50 µg/mL). In addition, the compound **8a** exhibited better antifungal activity against *R. solani* ($EC_{50}$ = 33.70 µg/mL) than hymexazol ($EC_{50}$ = 67.10 µg/mL). Our research found that the $R^1$ and $R^2$ groups of the title

compounds played a vital role in the antibacterial and antifungal activities. However, these studies can provide reference for similar chemical researches in the future. In the future, higher antimicrobial compounds against phytopathogenic microorganisms might be obtained via the further structural modification of 1,3,5-thiadiazine-2-thione derivatives.

### Funding
This work was supported by grants from the Fundamental Research Funds for the Central Universities of China (No. KYTZ201604 and KJSY201607) and the National Natural Science Foundation of China (No. 31772209). The funders had no role in study design, data collection and analysis, decision to publish, or preparation of the manuscript.

### Grant Disclosures
The following grant information was disclosed by the authors:
Fundamental Research Funds for the Central Universities of China: KYTZ201604 and KJSY201607.
National Natural Science Foundation of China: 31772209.

### Competing Interests
The authors declare that they have no competing interests.

### Author Contributions
- Jinghua Yan conceived and designed the experiments, performed the experiments, analyzed the data, prepared figures and/or tables, authored or reviewed drafts of the paper, approved the final draft.
- Weijie Si analyzed the data, authored or reviewed drafts of the paper, approved the final draft.
- Haoran Hu performed the experiments.
- Xu Zhao performed the experiments.
- Min Chen analyzed the data, contributed reagents/materials/analysis tools, authored or reviewed drafts of the paper, approved the final draft, analysis of single crystal.
- Xiaobin Wang analyzed the data, authored or reviewed drafts of the paper, approved the final draft.

### Data Availability
Crystallographic data are available in the Cambridge Crystallographic Data Centre: Jinghua Yan CCDC 1912576: Experimental Crystal Structure Determination, 2019, DOI 10.5517/ccdc.csd.cc22660j.

### Supplemental Information
Supplemental information for this article can be found online at http://dx.doi.org/10.7717/peerj.7581#supplemental-information.

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
