# Peer review of "Design, synthesis and antimicrobial activities of novel 1,3,5-thiadiazine-2-thione derivatives containing a 1,3,4-thiadiazole group"

_PeerJ, doi:10.7717/peerj.7581_

## Round 0.1 · original submission · Major Revisions

Dear Prof Yang

Enclosed are the review comments on your manuscript.
As you can see, one of the reviewers (R2) suggested its rejection. However, I could consider a revision if you are able to meet the suggestions and questions of Reviewer #2 (as well as Reviewer 1).

·

Basic reporting

There are a couple of misspelling in the manuscript, I corrected them and highlighted in the attached file.
References should be arranged in alphabetical order.
I believe there is a small mistake in stoichiometric calculation in experimental section
The new compounds are well characterized, however I missed a little more information or discusion about their synthesis.

Experimental design

No comments

Validity of the findings

No comment

Additional comments

The manuscript described a very interesting and well succeed hypothesis of mix two different classes of compound, with different biological activities in only one molecule to improve those biological activities, being possible to use a single compound for different application.
Nice work

Reviewer 2 ·

Basic reporting

The manuscript describes the design, the synthesis and the bioassay results of 1,3,5-thiadiazine-2-thione derivatives. The products synthesized demonstrated promise possibilities on the agrochemical field. However, an overview of the paper exhibits an insufficient discussion of the results, and an unclear findings about the improvements obtained in this paper compared with the literature.

Experimental design

The manuscript demonstrates a great number of results, and all compounds were proper characterized.

Validity of the findings

The manuscript demonstrates a poor discussion about the optimization of the synthetic procedures, other synthetic route available and yields variation. About the biological studies, they were conducted by structure modifications in a concise way, but without a deep and clear discussion.

Additional comments

So, I have added some comments that could improve the quality of the manuscript.

1- Materials – In this section, more information about the equipment and analysis should be provided: Model and probe of NMR; charge source and operating HRMS; and visualization procedures for TLC.
2- Line 181, putting data instead of date;
3- The Table 1 is presented in the Results and Discussion, but any discussion and/or arguments are described on the main text. Thus, I recommend putting the Table 1 in the support information. Additionally, I recommend using the standard models (organic synthesis) to describe all characterization data;
4- Small comments are described about the Table 2. Additionally, the points evaluated on the comments are not necessary because they are consolidated subjects on the base literature. I strongly recommend putting the Table 2 in the support information or inside of the protocols section, as described above, and a full discussion about synthetic route, purification and yields variation must be carried out. As the subtitle “Design and Synthesis”, a discussion about others synthetic protocols must be related;
5- I recommend putting the Table 5 in the support information. This discussion about the crystallographic data should be carried out before the biological evaluation and together with spectroscopy and spectrometry comments;
6- A figure with all structures evaluated on the biological assays should be added in the main text, before biological studies, for a clear understanding of the results;
7- The antibacterial effects of title compounds against Xoo should be added in a graphical form and not written in the main text;
8- During the biological evaluating a comparation with commercial and/or similar compounds should be carried out.

---

## Round 0.2 · accepted · Accept

The manuscript has been corrected and incremented according to the recommendations of the reviewers and the editor. It is suitable for publication in PeerJ now.